# Nutrients and Environmental Factors Cross Wavelet Analysis of River Yi in East China: A Multi-Scale Approach

**DOI:** 10.3390/ijerph20010496

**Published:** 2022-12-28

**Authors:** Lizhi Wang, Hongli Song, Juan An, Bin Dong, Xiyuan Wu, Yuanzhi Wu, Yun Wang, Bao Li, Qianjin Liu, Wanni Yu

**Affiliations:** Shandong Provincial Key Laboratory of Water and Soil Conservation and Environmental Protection, College of Resources and Environment, Linyi University, Linyi 276005, China

**Keywords:** environmental factors, nutrients, continuous wavelet transform, wavelet transform coherence

## Abstract

The accumulation of nutrients in rivers is a major cause of eutrophication, and the change in nutrient content is affected by a variety of factors. Taking the River Yi as an example, this study used wavelet analysis tools to examine the periodic changes in nutrients and environmental factors, as well as the relationship between nutrients and environmental factors. The results revealed that total phosphorus (TP), total nitrogen (TN), and ammonia nitrogen (${\mathrm{NH}}_{4}^{+}$–N) exhibit multiscale oscillation features, with the dominating periods of 16–17, 26, and 57–60 months. The continuous wavelet transform revealed periodic fluctuation laws on multiple scales between nutrients and several environmental factors. Wavelet transform coherence (WTC) was performed on nutrients and environmental factors, and the results showed that temperature and dissolved oxygen (DO) have a strong influence on nutrient concentration fluctuation. The WTC revealed a weak correlation between pH and TP. On a longer period, however, pH was positively correlated with TN. The flow was found to be positively correct with N and P, while N and P were found to be negatively correct with DO and electrical conductance (EC) at different scales. In most cases, TP was negatively correlated with 5-day biochemical oxygen demand (BOD_5_) and permanganate index (COD_Mn_). The correlation between TN and COD_Mn_ and BOD_5_ was limited, and no clear dominant phase emerged. In a nutshell, wavelet analysis revealed that water temperature, pH, DO, flow, EC, COD_Mn_, and BOD_5_ had a pronounced influence on nutrient concentration in the River Yi at different time scales. In the case of the combination of environmental factors, pH and DO play the largest role in determining nutrient concentration.

## 1. Introduction

Human activity has influenced the water status on the surface of the earth, resulting in water eutrophication, which is characterized by nitrogen (N) and phosphorus (P) pollutants [1]. As a result, an increasing amount of emphasis has been dedicated to the study of temporal and geographical variations in surface water quality [2,3]. Approximately 80% of the world’s lakes are controlled by P, 10% by P and N, and the rest by N or other factors [4]. As a result, N and P are important sources in the eutrophication of lake water quality. As highly bioactive elements, N and P are not only the primary source of lake eutrophication but also the limiting factor [5]. With the increase in the loading of N and P due to industrial emissions and agricultural fertilization, many shallow lakes and rivers have realized toxic algal blooms, which leads to the decline of aquatic biodiversity [6]. The temporal pattern characteristics of water quality in the basin can be found by evaluating long-term monitoring data of water N and P concentration as a critical index for further assessing the potential mechanisms influencing water quality evolution, managing water pollution in the basin, and maintaining water quality safety [7,8].

Nutrients, especially P, are considered the most important factors. There is a well-documented relationship between phytoplankton blooms and excess nutrient inputs into rivers [9]. Environmental factors may jointly regulate nutrients in complex ways. This highlights the need to simultaneously assess how variations in environmental factors interact with each other, concurrently influencing the nutrients [10]. One challenge with assessing interactions of environmental factors is articulating how these factors interact with each other and identifying their causal relationship with nutrients. Here, we used cross-wavelet analysis to explore their interactions by comparing the relationship between nutrients with the environmental factors in a multi-scale period. This may help water resource managers allocate resources in ways that will reduce the magnitude of eutrophication.

The classic multivariate statistical analysis method derives the correlation of surface water quality from water quality spatiotemporal monitoring data and then identifies the major pollution sources that cause spatiotemporal variation [11]. Multivariate statistical analysis, on the other hand, always presupposes that the data in time or the points in space are independent. Water quality time data has a particular continuity, and describing its similarity in time or place is insufficient [12,13]. As a result, approaches that can quantitatively quantify the time change pattern of water contamination must be used for research. Time series analysis encompasses both time–domain and frequency–domain analysis, which is a method for characterizing the change law of data. The time domain analysis approach obtains the data’s dynamic change law by describing the autocorrelation structure of the time series. This procedure is quite basic and straightforward. It is commonly utilized in time series studies of water quality [14]. Wavelet analysis is the most commonly used frequency domain analysis approach [15]. This method is mostly utilized in hydrology, geophysics, and other time series studies to characterize the periodic behavior or multiscale time pattern properties of signals, although its application in water quality time series is rare.

There is currently a scarcity of long-term water quality monitoring data in China, and there have been few reports on the spatiotemporal evolution analysis of surface water quality in the basin using time series analysis methodologies [16]. As a result, a comparative study to analyze the characteristics of water quality in the basin through time is quite valuable.

The River Yi was chosen as the case study region because it is the largest river originating in Shandong Province, East China, where both human activity and spatial variability in precipitation are considerable. According to Liu’s [17] multi-year surface water quality risk assessment of the River Yi basin, N and P pollution is the predominant pollution in the region. The time development of N and P pollution at each monitoring station is more complex and representative as the time scale changes. Furthermore, the impact of N and P on the reservoir basin’s surface water ecosystem requires greater attention.

This study examined the time series of total P (TP), total N (TN), and ammonia N (${\mathrm{NH}}_{4}^{+}$–N) concentrations in the River Yi (2005–2019), revealed the periodic pattern and time pattern characteristics of N and P pollutants using time–domain analysis and wavelet analysis. The periodic variations in nutrients and environmental indicators were analyzed using wavelet analysis methods in this study, which indicated the correlation between nutrients and environmental indicators. Therefore, this study aims to (i) analyze the multiscale oscillation features of nutrients (TP, TN, and ${\mathrm{NH}}_{4}^{+}$–N), and (ii) analyze and discuss the relationship between nutrients and environmental factors.

## 2. Materials and Methods

### 2.1. Study Area and Sampling

Water samples are collected monthly to monitor changes in environmental indicators of water. Sampling locations are evenly distributed in the top, middle, and lower portions of the River Yi. Figure 1 depicts the particular arrangement of sampling points.

The River Yi Basin has a mild temperate semi-humid monsoon continental climate that is influenced by both the continental air mass and the marine air mass. It has four different seasons, scorching summers, chilly winters, plenty of sunshine, and a long frost-free time. The basin’s annual average temperature is 13.05–14.3 °C. The annual sunlight hours are 1954–2535 h, with a large variance between north and south. The average annual precipitation in the basin is approximately 815 mm, and the rain is quite unequal, with the majority falling in the summer and autumn [18].

Every month, mixed water samples were collected from the upper layer of each sampling site (0, 0.5, and 1 m) using a Grasp sampler (Grasp BC–2300, Beijing, China). Following that, water samples were kept in a refrigerator at 4 °C for subsequent examination. Water samples were autoclaved at 121 °C for 30 min after the addition of K_2_S_2_O_8_ for TP and TN analyses. The molybdenum blue spectrophotometric method was used to measure the samples. The ${\mathrm{NH}}_{4}^{+}$–N concentration was determined using Nessler’s Reagent Colorimetry. The TP, TN, and ${\mathrm{NH}}_{4}^{+}$–N concentrations were determined using a continuous flow analyzer (Flowsys III, Systea Company, Zona Industriale Paduni-Selciatella, Italy). Shanghai N&D Co., Ltd. provided all of the materials for the analyses (Shanghai, China). All samples were examined in triplicate, and the results were expressed as the mean. The P concentration was determined according to previously described methods by Goulden and Brooksbank [19].

Two liters of water were collected to measure environmental factors [such as permanganate index (COD_Mn_) and five days biochemical oxygen demand (BOD_5_)]. BOD_5_ was measured with a BOD Trak^®^ II (Hach, Loveland, CO, USA), and COD_Mn_ using a COD_Mn_ III (Hach, USA). Before sampling in each site, the temperature, pH, and electrical conductance (EC) value were measured with a PHSJ–4A (Lei–ci, Shanghai, China), and the dissolved oxygen (DO) concentration was determined with a YSI 5750 (Yellow Springs, OH, USA). The flow velocity is measured using an ultrasonic Doppler flow meter (online type, Xiamen Boyida Technology Co., Ltd., Xiamen, China), and the flow is the monthly average.

### 2.2. Statistical Analysis

#### 2.2.1. Continuous Wavelet Transform

In earth sciences, the continuous wavelet transform (CWT) is a strong mathematical analysis tool. This transform decomposes time-series signals into a time-frequency space, inherits and develops the short-time Fourier transform’s localization property, and overcomes the window size’s inadaptability to frequency changes. Furthermore, it can give a time-frequency window whose size fluctuates with frequency and can deal with nonstationary signals more effectively [20,21]. The Morlet wavelet was chosen as the wavelet base function in this work; it is a non-orthogonal complex wavelet that represents a plane wave modulated by the Gaussian function [22]; it may be theoretically stated as:(1)$$\varphi_{0}(\eta )=\pi^{-1/4}e^{i\omega_{0}\eta }e^{-0.5\eta^{2}}$$

Here $i=\sqrt{-1}$, *ω*_0_ and *η* are the dimensionless frequency and space variables, respectively, whose corresponding values are set here to

*ω*_0_ = 6 and *η* = *s/x*. For the time series *X*1 of length *N*, its CWT form is
(2)$$W^{X1}(s,\tau )=\sqrt{\frac{\delta x}{s}}{\sum_{\tau =1}^{N}X_{1}}_{h}\psi \left[(h-\tau )\frac{\delta x}{s}\right]$$

Here $W^{X1}(s,\tau )$ denotes the wavelet transform coefficients of time series *X*_1_ with a scale factor s and a location *τ*, and *ψ*[] denotes the mother wavelet function.

#### 2.2.2. Wavelet Transform Coherence (WTC)

The WTC is used as a typical measure of the correlation between two time series. The WTC is defined mathematically as:(3)$$R_{n}^{2}=\frac{{\left|S\left(s^{-1}W_{i}^{XY}(s)\right)\right|}^{2}}{S\left(s^{-1}{\left|W_{i}^{X}(s)\right|}^{2}\right)\cdot S\left(s^{-1}{\left|W_{i}^{Y}(s)\right|}^{2}\right)}$$

Here *S* is a smoothing operator defined by the used wavelet type (the Morlet wavelet in this work), and
(4)$$W_{i}^{XY}(S)=W_{i}^{X}(S)\cdot W_{i}^{Y*}(S)$$

Here, the symbol * represents the complex conjugate. The statistical significance level against the noise background is estimated by the Monte Carlo simulation.

#### 2.2.3. Mapping and Mathematical Statistical Analysis

Wavelet analysis was performed using Matlab 2018 (Mathworks Co., Natick, MA, USA). ArcMap 10.8 was used to draw maps (https://www.esri.com/ (accessed on 1 March 2021)).

ANOVA was performed using SPSS v25.0 (IBM Co., Chicago, IL, USA). Differences were considered statistically significant at *p* < 0.05 and *p* < 0.01. Tukey test after one-way ANOVA was used to determine the differences between years and months of the water in the studied area.

## 3. Results

### 3.1. Yearly Variations in Water Nutrient Content

The TP content of the water was around five relatively high times. January–December 2005; January 2007–September 2009; July–December 2010; February 2012–March 2014; and January 2016–January 2017. The TP concentration of the water body varied dramatically across the years. The concentration of TP in the water body increased with time (Figure 2). The TP in 2006, 2011, 2014, and 2015 was significantly lower than that in the other years (*p* < 0.05). However, the TP in 2016 was significantly higher than that in the other years (*p* < 0.05).

The TN content of the water body changed in a relatively constant manner. With time, there was an overall increased tendency in TN content. The TN content in the water body increased noticeably between December 2012 and November 2014. Since 2016, the TN content has demonstrated a strong upward trend, ranging from 0.40 to 2.15 mg·L^−1^ (Figure 2). The TN in 2005 and 2006 was significantly lower than that in the other years (*p* < 0.05). However, the TN in 2013 was significantly higher than that in the other years (*p* < 0.05).

From 2005 until August 2010, the ${\mathrm{NH}}_{4}^{+}$–N content changed dramatically. From September 2010 until December 2015, the change was generally constant. After January 2016, the ${\mathrm{NH}}_{4}^{+}$–N content changed significantly. In general, ${\mathrm{NH}}_{4}^{+}$–N showed two periods of substantial variation and one phase of relative stability (Figure 2). ${\mathrm{NH}}_{4}^{+}$–N in 2006 was significantly lower than that in the other years (*p* < 0.05). However, the ${\mathrm{NH}}_{4}^{+}$–N in 2017 was significantly higher than that of the other years (*p* < 0.05).

Figure 3 depicted the monthly multi-year average change in nutrients in rivers in the research area. The content of TP began to rise in April and peaked in August. The TP content in the water body gradually decreased from September to March of the next year. As a result, throughout a one–year period, TP content changed in a single peak curve. The TP in August was significantly higher than that in April to June (*p* < 0.05). The TP in other months was not significantly different from each other (*p* > 0.05).

The average TN concentration in each month was rather steady, ranging between 0.5 and 1.5 mg·L^−1^. TN increased from December to April of the next year, notably in December. The TN content has greatly increased. The TN in different months was not significantly different from each other (*p* > 0.05).

The monthly average content variation of ${\mathrm{NH}}_{4}^{+}$–N was similar to that of TN, with a fluctuation range of 0.15–0.65 mg·L^−1^ most of the time. From January to April each year, the ${\mathrm{NH}}_{4}^{+}$–N concentration was generally high, and from May to December, it was relatively low. The ${\mathrm{NH}}_{4}^{+}$–N in January was significantly higher than that in September to November (*p* < 0.05). The ${\mathrm{NH}}_{4}^{+}$–N in other months was not significantly different from each other (*p* > 0.05).

### 3.2. Environmental Factors of Water in the River Yi

The changes in BOD_5_ and COD_Mn_ were similar. January to August 2005 and January to July 2016 were also relatively high periods. The rest of the time changes were very consistent. For most of the period, BOD_5_ ranged between 2.2 ± 0.2 and 4.0 ± 0.3 mg·L^−1^, while COD_Mn_ ranged between 2.6 ± 0.3 and 5.8 ± 0.3 mg·L^−1^. BOD_5_ and COD_Mn_ levels decreased with time (Figure 4). The BOD_5_ and COD_Mn_ in 2016 were significantly higher than that in the other years (*p* < 0.05).

The DO content was relatively high during two time periods: November 2009 to April 2012 and January 2017 to June 2019. During this period, the DO content fluctuated between 8.40 ± 0.21 and 14.56 ± 0.42 mg·L^−1^. Except for two particularly high times, DO content fluctuated reasonably steadily for the majority of the time, with a fluctuation range of 6.2–8.8 mg·L^−1^ (Figure 4). The DO in 2008 was significantly lower than that in the other years (*p* < 0.05).

EC remained very high from January 2005 to December 2006 (57.0 ± 5.1–61.0 ± 5.0 S·cm^−1^) and only declined significantly in October and November of 2005 and 2006 (38.0 ± 4.5–40.2 ± 4.0 S·cm^−1^). Since January 2007, EC suddenly plummeted, then showed a moderate increasing trend until September 2014, and since October 2014, EC has rapidly declined and then maintained a large variation (39.2 ± 4.3–63.0 ± 5.5 S·cm^−1^) (Figure 4). The EC in 2007 and 2008 was significantly lower than that in the other years (*p* < 0.05).

Every year, the rainy season in the research region lasted from July to August, and the abundant rainfall caused the River Yi’s flow to significantly grow. During the wet season, the River Yi’s flow could reach 300 ± 45–350 ± 50 m^3^·s^−1^. The highest observed flow rate was 805.4 ± 55 m^3^·s^−1^. The rest of the year was dry, and the average water flow ranged between 10 ± 5 and 30 ± 6 m^3^·s^−1^ (Figure 4). The flow in 2013 was significantly higher than that in the other years (*p* < 0.05).

Water temperature fluctuated seasonally. Summer temperatures were high, while winter temperatures were low. From June to September, the average surface water temperature was 28.4 ± 0.6 °C, and from December to March the following year, it was 2.1 ± 0.1 °C. Some river portions would freeze in the winter (Figure 4). The temperature in different years was not significantly different from each other (*p* > 0.05).

### 3.3. Periodic Change Characteristics of Nutrients

Wavelet analysis of nutrients in the River Yi was carried out to reveal the periodic law of nutrient changes in the River Yi. Figure 5 depicted the results. The contour map of the real part of wavelet coefficients can reflect periodic changes in the statistical series at different time scales and their distribution in the time domain, allowing the changing trend of the time series at different time scales to be judged. The real part value of the wavelet coefficient represents the signal strength. If the center of the contour line of the real part of the wavelet coefficient is positive, the nutrients are abundant at that time. If it is negative, it indicates that the nutrients are deficient at that time. The wavelet variance graph can be used to determine the dominant period in the evolution process by reflecting the distribution of fluctuation energy of nutrient time series with time scale.

TP showed 17, 26, and 57-month scale alternating changes. On the 57-month scale, the distribution law of extreme points is obvious. Throughout the study period, each period is relatively stable, with clear positive and negative phases. We discovered that the dominant period of TP in the River Yi basin was 17, 26, and 57 months (Figure 5a,b). From 2005 to 2019, these three periods had the highest levels of TP fluctuation energy in the River Yi. The 57–month time scale was the first peak, with the most significant periodic fluctuation energy.

The dominant period of ${\mathrm{NH}}_{4}^{+}$–N in the River Yi basin was 16 months (Figure 5c,d). The River Yi experienced 16, 26, and 60-month scale alternating changes. On the 60-month scale, the distribution law of extreme points was obvious. We discovered the dominant period of TN in the River Yi basin was 16, 26, and 60 months (Figure 5e,f). The 60-month scale was the first to peak, with the most significant periodic fluctuation energy.

To summarize, TP, TN, and ${\mathrm{NH}}_{4}^{+}$–N exhibited multi-scale oscillation with the dominant periods of 16–17, 26, and 57–60 months, respectively. In practical work, the period with too large a scale has no practical guiding significance. When combined with this study, the scale of 57–60 months was far too broad and has no practical application. As a result, this study only covers the 16–17 and 26–month periods.

### 3.4. Cross Wavelet Transform (XWT) of Nutrient’s Environmental Indexes

Cross wavelet analysis is a method of analyzing the correlation between two signals in the time-frequency domain that combines cross spectrum analysis and wavelet transform. Cross wavelet energy spectrums for monthly average nutrients and monthly average environmental indicators of the River Yi were calculated to analyze the multi-time scale correlation between nutrients and environmental indicators. The cross wavelet transform has excellent signal coupling and resolution properties. It can reveal two time series’ common high energy region and phase relationship. The phase vector’s right horizontal arrow indicates 0°, indicating that the corresponding environmental indexes of the nutrient peak rise together; the phase vector’s left horizontal arrow indicates 180°, indicating that the corresponding environmental indexes of the nutrient peak decrease together.

The cross wavelet transform (XWT) results of monthly average TP and monthly average environmental indexes revealed their interaction in different time and frequency domains, as well as the environmental indexes’ response characteristics to TP.

In 2006–2013 and 2016–2018, the interaction between TP and temperature was primarily concentrated in the dominant period of about 12 months. The phase angle was consistent, indicating that they have a significant correlation over 12 months (Figure 6a). The interaction between TP and pH was primarily concentrated in the dominant period of approximately 28 months in 2010–2012, indicating that they have a significant correlation in this period. For 28 months, the phase angle was 90 degrees. The interaction between the two was also visible in sub-periods of approximately two months and six months. Nonetheless, the duration of these subperiods was relatively short, ranging from half a year to a year (Figure 6b). The interaction between TP, flow, and EC was low, there was no obvious dominant period, and the period was only about 12 months around 2017. The phase angle shifted dramatically (Figure 6c,d). The interaction between TP and DO was primarily concentrated on the 12-month dominant period in 2009–2012 and 2016–2018 (Figure 6e). The dominant period’s phase angle was the opposite, indicating that they have a significant correlation over 12 months (Figure 6e). There was little interaction between TP, COD_Mn_, and BOD_5_. There was no obvious dominant period. There was only a period of 16 months between TP and COD_Mn_ in 2016–2018 (Figure 6c,d). The phase angle shifted dramatically (Figure 6f,g).

In 2011–2016 and 2018–2019, the interaction between TN and temperature was primarily concentrated in the dominant period of about 12 months, indicating that they have a significant correlation in the period of 12 months (Figure 6h). The opposite phase angle showed that the TN correlated with the temperature peak decreases (Figure 6h). The interaction of TN and pH was minimal. There was no discernible time frame (Figure 6i). The interaction between TN and flow was primarily concentrated on the 8-month dominant period in 2013–2014. Both have the same phase angle, indicating that they have a significant correlation over eight months (Figure 6j). There was little interaction between TN and Ec. No obvious dominant period was observed between TN and EC (Figure 6k). In 2012–2015 and 2018–2019, the interaction between TN and DO was primarily concentrated on the dominant period of about 12 months, indicating that they have a significant correlation in the period of 12 months (Figure 6l). Both have the same phase angle, indicating that they have a significant correlation in the 12-month period (Figure 6l). The interaction of TN, COD_Mn_, and BOD_5_ was minimal. There was no obvious dominant period (Figure 6m,n). In 2013–2015, there was only an 8-month period. The phase angle shifted dramatically (Figure 6m,n). The phase angles of TN and COD_Mn_, as well as TN and BOD_5_, were opposed. It meant that the TN corresponding to the COD_Mn_ and BOD_5_ peaks was decreasing.

In 2008–2019, the interaction between ${\mathrm{NH}}_{4}^{+}$–N and the temperature was primarily concentrated in the dominant period of 12 months, indicating that they have a significant correlation in the 12-month period (Figure 6o). ${\mathrm{NH}}_{4}^{+}$–N and temperature have opposite phase angles from 2008 to 2011 and 2014 to 2019, indicating that ${\mathrm{NH}}_{4}^{+}$–N corresponds to temperature peak decreases. From 2011 to 2014, ${\mathrm{NH}}_{4}^{+}$–N and temperature have the same phase angle, indicating that they have a significant correlation in the period of 12 months (Figure 6o). The interaction of ${\mathrm{NH}}_{4}^{+}$–N and pH was minimal. There was no obvious dominant period (Figure 6p). The interaction between ${\mathrm{NH}}_{4}^{+}$–N and flow was primarily concentrated on the 12-month period in 2016–2018; both have opposite phase angles, indicating that ${\mathrm{NH}}_{4}^{+}$–N corresponds to flow peak decreases (Figure 6q). The interaction between ${\mathrm{NH}}_{4}^{+}$–N and EC was primarily concentrated on the dominant period of 12 months in 2015–2017; both have opposite phase angles, indicating that ${\mathrm{NH}}_{4}^{+}$–N corresponds to the EC peak decreases (Figure 6r). The interaction between ${\mathrm{NH}}_{4}^{+}$–N and DO was primarily concentrated on the dominant period of 12 months in 2009, 2013, and 2015–2019; both have the same phase angles, indicating a significant correlation in the dominant period of 12 months (Figure 6s). There was little interaction between ${\mathrm{NH}}_{4}^{+}$–N, COD_Mn_, and BOD_5_; there was no clear dominant period (Figure 6t,u). In 2015–2017, there was only a 12-month period. The phase angles of ${\mathrm{NH}}_{4}^{+}$–N, COD_Mn_, and BOD_5_ were the same, indicating a significant correlation in the period of 12 months (Figure 6t,u).

### 3.5. Wavelet Transforms Coherence of Nutrients and Environmental Indexes

The cross wavelet coherence spectrum compensated for the cross wavelet spectrum’s shortcoming in correlation analysis in the low energy range. The cross wavelet coherence spectrum of nutrients and environmental markers was depicted in Figure 7 (The isoline in the figure was the square of the wavelet correlation coefficient). The cross wavelet coherence spectrum has a larger time-frequency domain space than the cross wavelet spectrum.

TP was positively correlated with temperature in the 12-month period from 2009 to 2014 (Figure 7a). TP was positively correlated with pH from 2009 to 2013 (24-month period) and from 2018 to 2019 (5-month period). TP was negatively correlated with pH from 2007 to 2014 and from 2018 to 2019 (5-month period) (Figure 7b). TP and flow were weakly correlated at different time scales. Only in the 3-month period of 2010, there was a positive correlation between TP and flow (Figure 7c). TP and EC were weakly correlated at different time scales. Only in the 4-month period of 2005 and 2018–2019 was there a positive correlation. TP was negatively correlated with EC in the 18-month period from 2007 to 2008 (Figure 7d). TP was negatively correlated with DO in different periods. In the 12-month period of 2009 and 2012–2014, there was a negative correlation between TP and DO (Figure 7e). TP is positively correlated with COD_Mn_ in the 6-month period from 2006 to 2008 (Figure 7f), and is negatively correlated with COD_Mn_ in the 6-month period from 2009 to 2012. TP and COD_Mn_ were significantly correlated in the period of 16 months from 2014 to 2018, with a phase angle of 90°. When the scale was greater than 32 months, TP and COD_Mn_ were positively correlated. However, it has no practical meaning, because the time scale is too large (Figure 7f). TP was negatively correlated with BOD_5_ in 12 month periods from 2009 to 2012 (Figure 7g). In a small-scale period of 3 months, TP and BOD_5_ have multiple related areas. When the scale was greater than 30 months, TP and BOD_5_ were positively correlated, but because the time scale was too large, it has no practical meaning (Figure 7g).

TN was negatively correlated with temperature in the 20-month period from 2011 to 2016 and the 6-month secondary period of 2014 and 2016–2018 (Figure 7h). TN was positively correlated with pH from 2009 to 2011 (an 8-month period), and the correlation was weak in other periods (Figure 7i). TN was positively correlated with flow from 2013 to 2016 (1–8-month period). The results showed that the flow has a significant impact on TN content in different periods from 2013 to 2016 (Figure 7j). TN was negatively correlated with EC in the 18-month period from 2010 to 2016 and the 6-month secondary period of 2014 (Figure 7k). The correlation between TN and DO was weak, and there was no apparent dominant period. TN was positively correlated with DO from 2009 to 2010 (7-month period), 2012–2013 (5-month period), and negatively correlated with DO from 2013 to 2015 (5-month period) (Figure 7l). The correlation between TN and COD_Mn_, BOD_5_ was weak, and there was no apparent dominant period (Figure 7m,n).

${\mathrm{NH}}_{4}^{+}$–N was negatively correlated with temperature in the 12-month period from 2014 to 2018 and the 6-month secondary period of 2016 (Figure 7o). ${\mathrm{NH}}_{4}^{+}$–N was positively correlated with temperature in the 6-month secondary period of 2008 and 2014 (Figure 7o). ${\mathrm{NH}}_{4}^{+}$–N was positively correlated with pH in the 24-month period from 2009 to 2014, and in the 8-month secondary period from 2015 to 2018 (Figure 7p). The correlation between flow and ${\mathrm{NH}}_{4}^{+}$–N was weak. ${\mathrm{NH}}_{4}^{+}$–N was negatively correlated with the flow in the approximately 4-month period in 2014. ${\mathrm{NH}}_{4}^{+}$–N was correlated with the flow in the about 6-month period in 2010, with a phase angle of 270° (Figure 7q). ${\mathrm{NH}}_{4}^{+}$–N was negatively correlated with EC in the 12-month period from 2014 to 2018, and the 4-month secondary period from 2012 to 2013. ${\mathrm{NH}}_{4}^{+}$–N was positively correlated with EC in the 4-month secondary period of 2014 (Figure 7o). ${\mathrm{NH}}_{4}^{+}$–N was correlated with EC in the 7-month period in 2010, with a phase angle of 90° (Figure 7r). The correlation between DO and ${\mathrm{NH}}_{4}^{+}$–N was weak. ${\mathrm{NH}}_{4}^{+}$–N was positively correlated with DO in the 5-month period from 2008 to 2009 (Figure 7s). ${\mathrm{NH}}_{4}^{+}$–N was correlated with COD_Mn_ in about 5-month period from 2008 to 2010, with a phase angle of 270°, and in the 7-month period of 2012, with a phase angle of 90° (Figure 7t). ${\mathrm{NH}}_{4}^{+}$–N was positively correlated with BOD_5_ in the about 15-month period from 2013 to 2018, and in the 5-month secondary period from 2011 to 2012 and 2017 (Figure 7u).

## 4. Discussion

Many factors influence the concentration of nutrients such as N and P in water, including environmental conditions such as temperature, DO, pH, disturbance, light, and so on. When environmental parameters (such as temperature, DO, pH, and so on) are favorable for eutrophic water bodies, N and P in sediments can be released, causing the concentration of N and P in water bodies to grow or be maintained at a higher level [23,24]. As a result, it is critical to investigate the correlation between environmental conditions and N and P in water bodies.

### 4.1. Effect of Temperature on the Concentrations of N and P in Water

The temperature change follows the seasonal change rule. There was a 12-month (one-year) periodic change rule between ${\mathrm{NH}}_{4}^{+}$–N, TN, TP, and temperature in the River Yi in this study, therefore the N and P content in the River Yi’s water has an apparent interannual change rule. The nutrients in the water increased in the summer and decreased in the winter as the temperature of the water changed.

The effect of temperature on N and P exchange between sediment and water mostly affects the rate of organic P and organic N mineralization. It has an indirect impact on the release of N and P in sediment [25]. The P release of sediment will increase dramatically when the temperature rises [26,27]. Liu et al. [17] showed that the sediment might release N and P during the summer. This was mostly due to higher summer temperatures and the activity of microorganisms in the sediment [28]. On the one hand, it accelerated the process of biological disturbance and mineralization; while lowering DO concentration in water accelerated anaerobic transformation [29]. The presence of these two conditions promoted the transformation and release of organic P and organic N in sediments [30,31]. Our results are in agreement with these previous studies. This should be a significant explanation for the seasonal change in nutrient concentration.

Simultaneously, because summer is the rainy season in the research area, a considerable number of pollutants flow into the river from surface runoff, increasing N and P concentrations in the River Yi. Furthermore, aquatic plants, particularly *Potamogeton crispus*, which propagates in large numbers in the river, decay in the summer and release a large amount of N and P into the water, causing the N and P content in the water to increase, and causing the N and P content in the water body to change periodically with temperature.

### 4.2. Effect of pH on N and P Concentration in Water

The periodicity between pH and N and P is not visible in this investigation, although there is a correlation on a longer scale period (Figure 6b,i,p). The pH value of river water is an essential indicator of water quality. pH value directly impacts N and P release in sediments by acting on the redox and replacement interactions between various ions at the sediment-water interface. The amount of N and P released from sediments rises dramatically under acidic and alkaline conditions, with the least amount released under neutral conditions [32]. On a long-time scale, TP and TN were found to be positively correlated with pH in this study (Figure 7b,i,p). It demonstrated that the quantity of N and P in water bodies increased with increasing pH value over a long period, which may be related to river sediment release and removal. The pH change range of the water body in this study is primarily 7.5–8.0, and the water body is weakly alkaline (Figure 4). When the pH of water is alkaline, OH^−^ ions interact with H_2_PO_4_^−^ in the sediment, stimulating the release of NaOH-P (ferrous and aluminum P) in the sediment. P in water exists in the form of H_2_PO_4_^−^ and HPO_4_^2−^ when pH is neutral; under neutral conditions, Al^3+^ will hydrolyze to produce colloids, which will promote P adsorption in water and form Al-P to deposit in sediment [33]. In this study, this should be the main mechanism of interaction between P concentration and pH on a large scale.

### 4.3. Effect of Flow on N and P Concentration in Water

Flow is a key component influencing the N and P cycle in river sediment–water interfaces, and flow has a direct impact on N and P release in sediments. Water movement can supply nutrients and water flow can produce water disruption, both of which have an impact on N and P exchange at the water-sediment interface [34]. Liu et al. [35] have shown that disturbance alters the amount of distribution of N and P in distinct forms in sediments.

Disruption has two major effects on the release of N and P from sediments. First, disturbance accelerates the diffusion process and directly accelerates the migration rate of N and P to the direction of low-concentration overlying water; Second, disturbance promotes N and P release from sediment to overlying water through particulate N and P resuspension. According to Omogbehin et al. [36], the higher the disturbance rate, the more N and P are released from the sediment.

The flow of the River Yi water increases rapidly with the beginning of the rainy season in July (Figure 4). The disruption of the water body and the pollutants delivered upstream are the primary causes of the rapid increase in N and P content in the water. The seasonal fluctuation in the River Yi’s flow causes a seasonal change in the nutrition. This is the most likely explanation for the positive correlation between flow and N and P at different scales.

There was a 12-month interval between flow and N, P in this study. The key factor influencing the N and P content of the water body was interannual flow variation. At the same time, the cross wavelet coherence spectrum revealed that flow was most of the time positively correlated with N and P, implying that the N and P content in the water rose as the flow increased.

### 4.4. Effect of EC on N and P Concentration in Water

EC is an important environmental factor that influences the release of N and P from sediment into the overlying water. EC changes cause a shift in redox potential, which controls the redistribution of N and P in these components, resulting in the changes of N and P concentrations in the water body and its eutrophication potential. When EC is low, the sediment has a critical N and P release capacity [37]. When the sediment is at low EC, the absolute concentration of active inorganic N and P increases, and it is simpler to transition into unstable N and P forms under low redox conditions, resulting in an increase in N and P content in the water body [38]. The EC in 2007 and 2008 was significantly lower than that in the other years (Figure 4). This is the main reason for the study’s negative correlation between EC and N and P in this period. This is also the main reason for the negative correlation between EC and TN and ${\mathrm{NH}}_{4}^{+}$–N.

### 4.5. Effect of DO on N and P Concentration in Water

The change in DO content will result in a change in redox conditions at the sediment–water interface, which will result in the transformation of the forms of Fe, Al, and their bound N and P, affecting the adsorption and release of N and P in the sediment [39]. In general, when the DO content is low (anoxic), the sediment releases N and P [40]. The main mechanism is that the oxides or hydroxides of Fe (III) in the sediment are reduced to oxides or ferric hydroxides of Fe (II), allowing the N and P absorbed by the oxides or hydroxides of Fe (II) to be easily released into the interstitial water [41]; on the contrary, it shows N and P adsorption. Temperature variation is frequently correlated with P release in sediments. The explanation for this could be that rising temperatures stimulate the activity of microbes and enzymes in sediments, increasing the rate of organic matter decomposition [42,43].

The DO in the research region increases in the spring owing to *P. crispus* reproduction and decreases in the summer due to *P. crispus* decline in the water body. *P. crispus* growth may also absorb N and P from the River Yi, resulting in a decrease of N and P in the water [44]. N and P are released into the water during the decomposition of *P. crispus*, which is a major reason influencing the periodic variation in N and P concentration in the water. This is the primary reason for the study’s negative correlation between DO and N and P at different scales.

### 4.6. Effect of COD_Mn_ and BOD_5_ on N and P Concentration in Water

In general, the water body’s self-purification capacity is greater in the wet season than in the dry season, but the concentration of COD_Mn_ and BOD_5_ increases in the wet season, indicating that organic pollution along the River Yi is relatively severe, and a large number of organic pollutants enter the water body via surface runoff during the wet season [45]. Jarvie et al. [46] investigated the N and P in certain rivers on England’s east coast and discovered a substantial negative correlation between the N and P and the flow in some rivers. This was primarily due to the coastal areas of these rivers being more affected by point source pollution than non-point source pollution, and increased runoff has a dilution effect on pollutants; however, some other rivers showed a positive correlation, owing to non-point source pollution being the primary pollution mode in this area [47].

COD_Mn_ and BOD_5_ concentrations in this research region tend to rise in the summer and fall in the winter, which may be related to non-point source pollution upstream in the summer. Summer precipitation carried upstream contaminants into the River Yi, resulting in an increase in COD_Mn_ and BOD_5_ concentration. As a result of the seasonal variable correlation between COD_Mn_ and BOD_5_, and flow in the River Yi’s water body, organic pollution along the River Yi is primarily non-point source pollution.

## 5. Conclusions

Wavelet analysis revealed that TN, TP, and ${\mathrm{NH}}_{4}^{+}$–N exhibit multiscale oscillation features, with dominating periods of 16–17, 26, and 57–60 months, respectively.

The periodic change rule between ${\mathrm{NH}}_{4}^{+}$–N, TN, TP, and temperature in the water body was 12 months (one year). The seasonal temperature change caused environmental elements in the water body to vary, which was the primary cause of the periodic shift of N and P in the water body.

The periodicity between pH and ${\mathrm{NH}}_{4}^{+}$–N, TN, and TP was not apparent, but pH and TN were positively correlated in the long-time scale.

The flow was positively correct with N and P at different scales. There was a 12-month period between flow and N, P. The N and P content in the water increased with the increase in flow. DO and EC were negatively correct with N and P at different scales. TP was negatively correlated with BOD_5_ and COD_Mn_ for most of the period. The correlation between TN and COD_Mn_, and BOD_5_ is weak, and there is no apparent dominant period.

Starting with the interaction between nutrients and environmental factors in the River Yi, this study investigated the multi-scale spectrum of changes in environmental factors and nutrient content, as well as the implications of these changes. The river is a massive water container. The water cycle process includes not only the balance of water quality, but also the conversion of energy, the geographical environment of the reservoir area, the world climate, and other complicated factors. As a result, a thorough examination of the effects of numerous factors on nutrition is required. To examine and discuss the interaction between nutrients and environmental conditions, the cross wavelet transform method is used. Although it is a statistical relationship, it might disclose a deeper level of intrinsic interaction between many factors. The cross wavelet transform method is a powerful physical mechanism-based tool for studying the link between nutrients and environmental factors.

## Figures and Tables

**Figure 1 ijerph-20-00496-f001:**
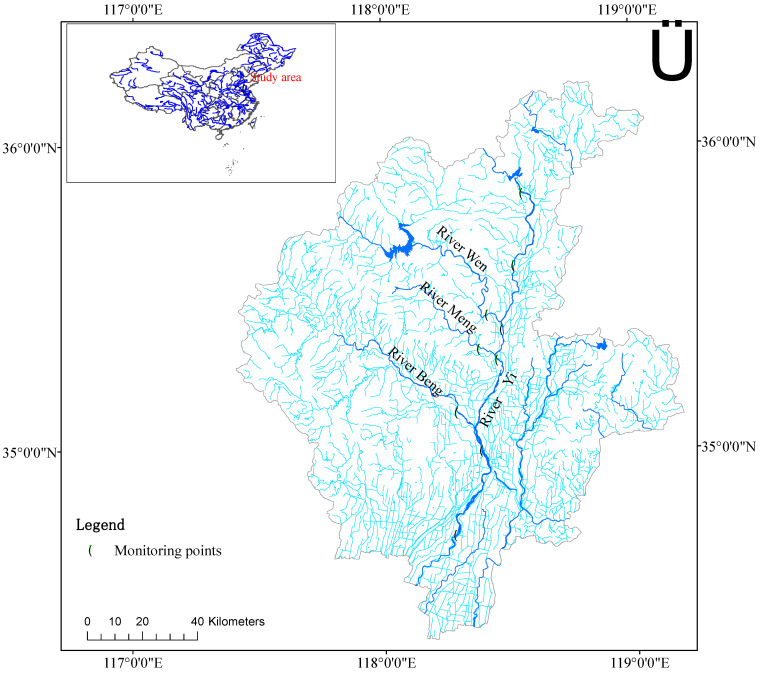
Study region and sampling locations. ArcMap version 10.8 (https://www.esri.com/ (accessed on 1 March 2021)) was used to create the map.

**Figure 2 ijerph-20-00496-f002:**
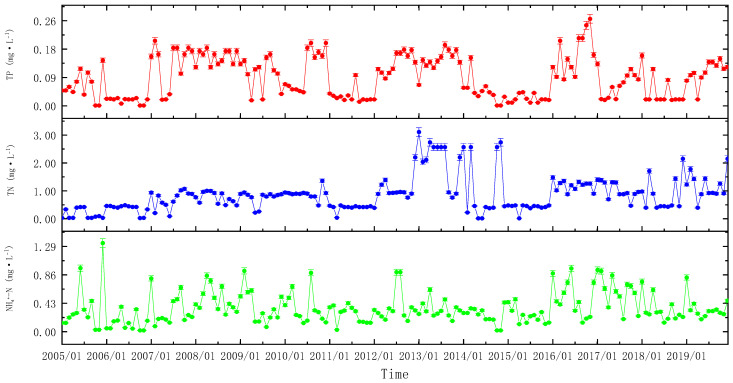
Changes in N and P nutrients in the River Yi from 2005 to 2019 (means ± SD).

**Figure 3 ijerph-20-00496-f003:**
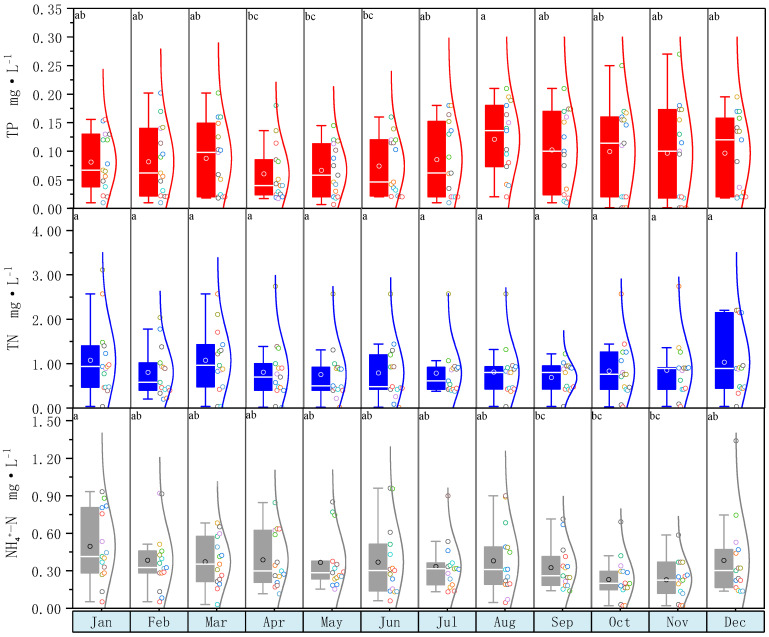
River Yi average monthly nutrient content for 2005–2019, different lowercase letters indicate significant differences between nutrients among different months at 0.05 significant level.

**Figure 4 ijerph-20-00496-f004:**
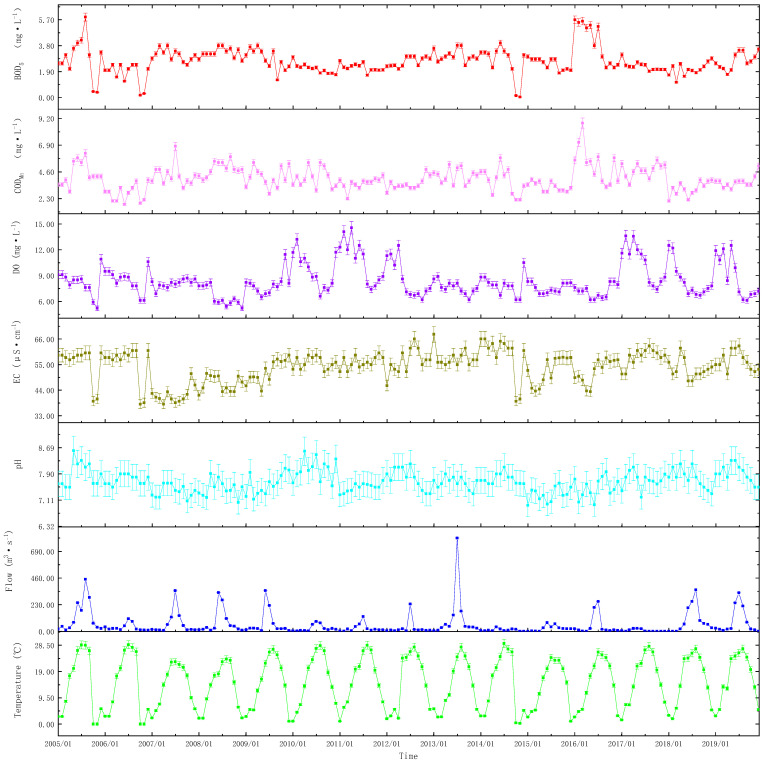
Changes in environmental factors in the River Yi from 2005 to 2019 (means ± SD).

**Figure 5 ijerph-20-00496-f005:**
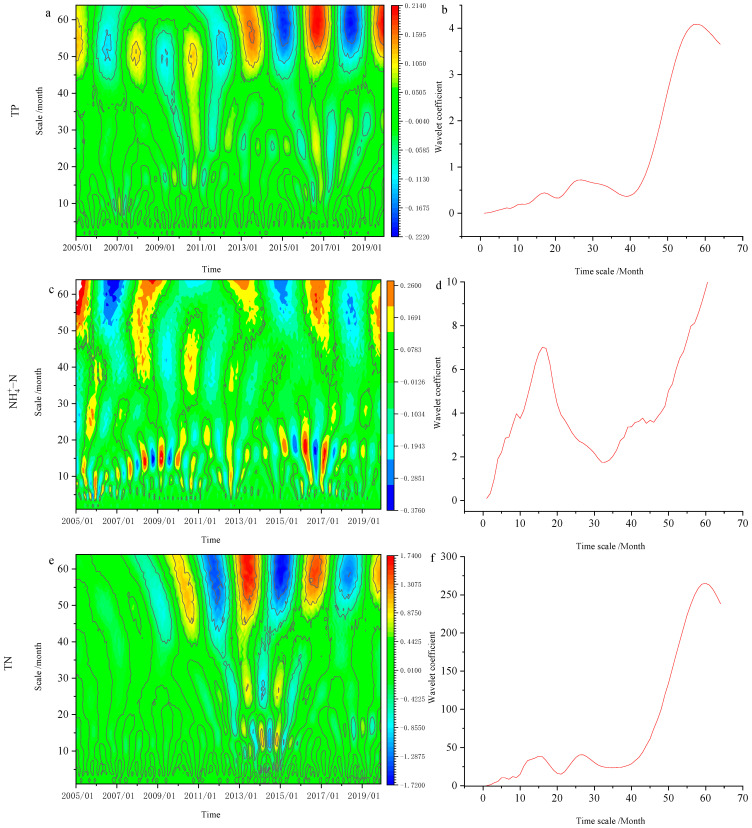
Real part isoline of wavelet coefficients and wavelet coefficient variance of nutrients in the River Yi. (**a**,**c**,**e**) are real part isoline of TP, ${\mathrm{NH}}_{4}^{+}$–N, TN. (**b**,**d**,**f**) are the dominant period of TP, ${\mathrm{NH}}_{4}^{+}$–N, TN.

**Figure 6 ijerph-20-00496-f006:**
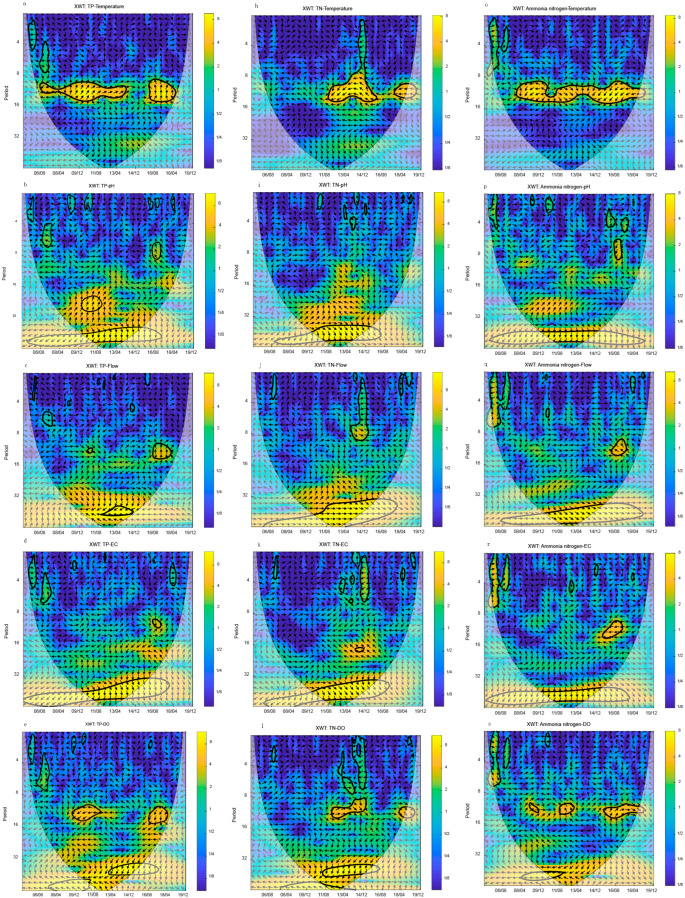
Cross wavelet transform (XWT) of the nutrient concentration. The thick black contour indicates a 95% confidence level against yellow noise and the cone of influence is shown as the lighter shade. The arrows (vectors) designate the phase difference (within–phase pointing right, anti-phase pointing left). The arrow to the left indicates negative phase coherence, and again the darker the color, the stronger the negative coherence. *Y*-axis period is month, the *X*-axis date format is year/month. (**a**–**g**) are XWT of TP and Temperature, pH, Flow, EC, DO, COD_Mn_, and BOD_5_. (**h**–**n**) are XWT of TN and Temperature, pH, Flow, EC, DO, COD_Mn_, and BOD_5_. (**o**–**u**) are XWT of ${\mathrm{NH}}_{4}^{+}$–N and Temperature, pH, Flow, EC, DO, COD_Mn_, and BOD_5_.

**Figure 7 ijerph-20-00496-f007:**
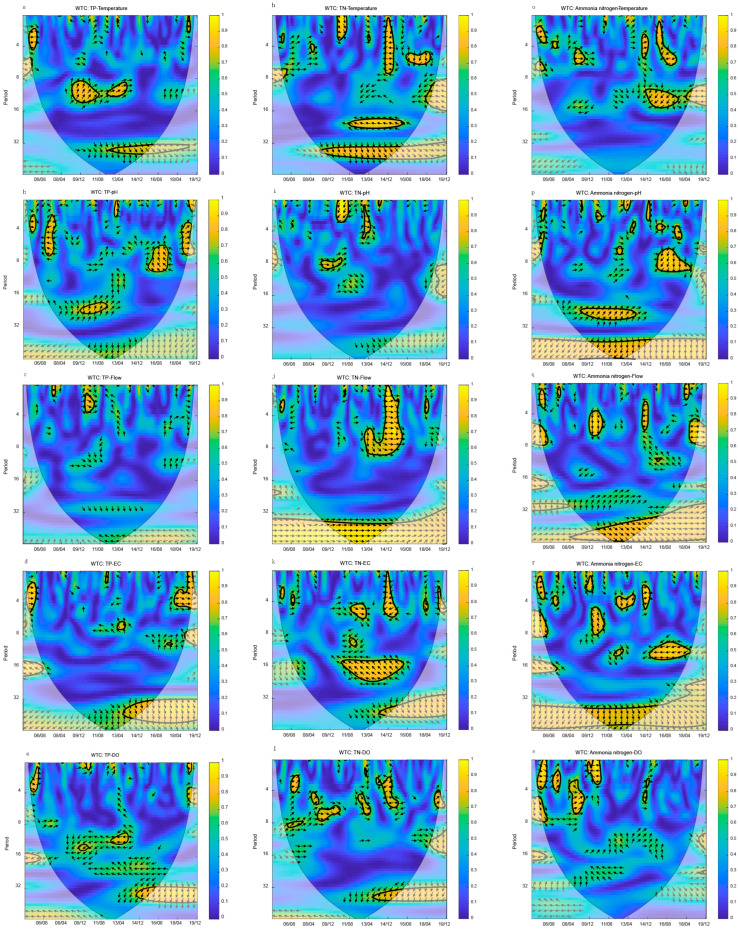
Wavelet transform coherence of the nutrient concentration. The thick black contour indicates a 95% confidence level against yellow noise and the cone of influence is shown as the lighter shade. The arrows (vectors) designate the phase difference (within-phase pointing right, anti-phase pointing left). The arrow to the left indicates negative phase coherence, and again the darker the color, the stronger the negative coherence. *Y*-axis period is month, the *X*-axis date format is year/month. (**a**–**g**) are WTC of TP and Temperature, pH, Flow, EC, DO, COD_Mn_, and BOD_5_. (**h**–**n**) are WTC of TN and Temperature, pH, Flow, EC, DO, COD_Mn_, and BOD_5_. (**o**–**u**) are WTC of ${\mathrm{NH}}_{4}^{+}$–N and Temperature, pH, Flow, EC, DO, COD_Mn_, and BOD_5_.

## Data Availability

The data used to support the findings of this study are available from the corresponding author upon request (e-mail: wanglizhi@lyu.edu.cn and anjuan0715@126.com).

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
