# Peer review of "Nutrients and Environmental Factors Cross Wavelet Analysis of River Yi in East China: A Multi-Scale Approach"

_ijerph, 2022, doi:10.3390/ijerph20010496_

Round 1
Reviewer 1 Report
Dear colleagues,
Thank you for the opportunity to read your manuscript, which discusses a current topic, namely the use of wavelet cross-analysis tools to examine periodic changes in nutrients and physical and chemical indicators, as well as the relationship between nutrients and physical and chemical indicators.
What can you tell me about the research gap? What does this article bring to the specialized literature? the answer to this question would increase the quality of the manuscript.
Sincerely,
Author Response
We feel great thanks for your professional review work on our article. As you are concerned, there are several problems that need to be addressed. According to your nice suggestions we have made extensive corrections to our previous draft. The detailed corrections are listed below.
Thank you for the opportunity to read your manuscript, which discusses a current topic, namely the use of wavelet cross-analysis tools to examine periodic changes in nutrients and physical and chemical indicators, as well as the relationship between nutrients and physical and chemical indicators.
What can you tell me about the research gap? What does this article bring to the specialized literature? the answer to this question would increase the quality of the manuscript.
We sincerely appreciate your insightful comments. We revised this section in response to the reviewer's suggestions. The discussion of relevant contents has been included to the introduction and conclusion.
Reviewer 2 Report
The present manuscript entitled “Multi–scale characteristics of the Yi River (East China) between nutrients and environmental factors based on cross wavelet analysis” is a quality research conducted by the authors highlighting the importance of nutrient accumulation in the eutrophication of the water that results in damage to water quality. I appreciated the hard work and research concerns of the authors. However, despite the importance of the idea, the manuscript needs to be thoroughly checked grammatically, although I am not native to English. Moreover, I highlighted some changes in different sections of the manuscript, which may further aid in the quality of the manuscript. I will recommend major revisions as most of the changes are changes are of concerned and some are topographic mistakes which will further improve quality of the manuscript.
Title: The title is very confusing need to be revised for understanding of the readers and must also be attractive one like “Nutrients and Environmental factors cross wavelet analysis of Yi River in East China: A multi-scale approach’
The abstract is written in understandable language, however, few minor correction are needed
- Line 13. Nutrient, Physical and chemical indicator.. I feel nutrients are also chemical indicator so please include it in chemical indicator if possible..
-Line 16-17. Incomplete sentence
-Line 18. Delete “both”
-Line 22. BOD5. Did you mean five days biochemical oxygen demand?
-CODMn..Please clarify Mn, Is it permanganate or manganese??
-Line 22-25. Proper comprehensive conclusion is missing
Introduction
The introduction is understandable although the sentence structure can be improved. However, the author has analyzed variety of nutrients and chemicals but in introduction emphasizes only Nitrogen and Phosphorus. It needs to be improved and literature needs to be added highlighting the importance of all the chemical and nutrients and factors included in the manuscript. In addition, the importance of eutrophication and its impacts on water quality if added in introduction, it will improve the subject matter and will aid to the importance of the study. Some specific comments are as follow
-Line 31. Amount of emphasis.. Very vogue please revises
-Line 32. Correct the citation please and check for others also
-Proper hypothesis and objective needed to be formulated
Methodology
Well and comprehensive written but need topographic correction.
-Line 90. Please remove “depicts the”
Results
The results are interesting but have some minor corrections and suggestion which are as follow.
-Line 145. Revise please very vogue
-I will suggest using some statistical analysis like ANOVA to evaluate the change in nutrient concentration with time, such statistical tool will better revealed the temporal changes in nutrients concentration (figure 2)
-same I will suggest for Figure 3 to add the P and F-value
-Line 183. Ec may be replaced with EC
-Figure 4. Also need to statistically evaluated the differences.. Moreover Mean±SE ..But I have not seen any such value
Discussion
The discussion in written in good language but it’s not coherent with results. The result revealed the temporal changes of nutrients and chemical parameters. In the methodology physical factors are included but, in result the physical factors i.e. pH, Electrical conductivity is discussed, however, temperature data are missing and in the discussion, the changes in nutrients were linked with temperature. This need to be clarified, if temperature variations are considered for season than it needs to be evaluated. Some other minor corrections are as follow
-The section need to be checked for spelling and grammar mistakes
Conclusion
The conclusion seems like summery of the results, it don’t provide further guidelines for study. If possible provide assessment about the outcomes and uses of the current study for future and also suggestion recommendation based on current findings.
Author Response
We feel great thanks for your professional review work on our article. As you are concerned, there are several problems that need to be addressed. According to your nice suggestions we have made extensive corrections to our previous draft. The detailed corrections are listed below.
The present manuscript entitled “Multi–scale characteristics of the Yi River (East China) between nutrients and environmental factors based on cross wavelet analysis” is a quality research conducted by the authors highlighting the importance of nutrient accumulation in the eutrophication of the water that results in damage to water quality. I appreciated the hard work and research concerns of the authors. However, despite the importance of the idea, the manuscript needs to be thoroughly checked grammatically, although I am not native to English. Moreover, I highlighted some changes in different sections of the manuscript, which may further aid in the quality of the manuscript. I will recommend major revisions as most of the changes are changes are of concerned and some are topographic mistakes which will further improve quality of the manuscript.
Title: The title is very confusing need to be revised for understanding of the readers and must also be attractive one like “Nutrients and Environmental factors cross wavelet analysis of Yi River in East China: A multi-scale approach’
We think this is an excellent suggestion. We have modified the title according to the reviewer’s suggestion.
The abstract is written in understandable language, however, few minor correction are needed
- Line 13. Nutrient, Physical and chemical indicator.. I feel nutrients are also chemical indicator so please include it in chemical indicator if possible..
In this paper, physical and chemical indicators actually refer to environmental factors, so they have been modified throughout this paper.
-Line 16-17. Incomplete sentence
We think this is an excellent suggestion. We have re-written this part according to the reviewer’s suggestion. The sentence has been modified according to the context
-Line 18. Delete “both”
We were really sorry for our careless mistakes. Thank you for your reminder. The word “both” has been deleted
-Line 22. BOD5. Did you mean five days biochemical oxygen demand?
Yes, the article has revised the relevant statement
-CODMn..Please clarify Mn, Is it permanganate or manganese??
CODMn is expressed as permanganate index
-Line 22-25. Proper comprehensive conclusion is missing
comprehensive conclusion has been added
Introduction
The introduction is understandable although the sentence structure can be improved. However, the author has analyzed variety of nutrients and chemicals but in introduction emphasizes only Nitrogen and Phosphorus. It needs to be improved and literature needs to be added highlighting the importance of all the chemical and nutrients and factors included in the manuscript. In addition, the importance of eutrophication and its impacts on water quality if added in introduction, it will improve the subject matter and will aid to the importance of the study. Some specific comments are as follow
We think this is an excellent suggestion. We have revised this part according to the reviewer suggestion.
-Line 31. Amount of emphasis.. Very vogue please revises
This sentence has been revised according to expert opinions
-Line 32. Correct the citation please and check for others also
We sincerely appreciate the valuable comments. We have checked the literature carefully and added more references into the literature in the revised manuscript
-Proper hypothesis and objective needed to be formulated
Proper hypothesis and objective have been corrected.
Methodology
Well and comprehensive written but need topographic correction.
-Line 90. Please remove “depicts the”
We were really sorry for our careless mistakes. Thank you for your reminder. The word “depicts the” has been deleted
Results
The results are interesting but have some minor corrections and suggestion which are as follow.
-Line 145. Revise please very vogue
-I will suggest using some statistical analysis like ANOVA to evaluate the change in nutrient concentration with time, such statistical tool will better revealed the temporal changes in nutrients concentration (figure 2)
We sincerely appreciate the valuable comments. We have revised this part according to the reviewer's suggestion.
-same I will suggest for Figure 3 to add the P and F-value
Because adding P and F-value to the diagram will make the diagram more complex and difficult to interpret, this paper uses the method of adding lowercase letters to express differences.
-Line 183. Ec may be replaced with EC
We sincerely appreciate the valuable comments. It has been modified here, and it has been modified in the full text, and the expressions in all figures have also been modified according to the reviewer's suggestion.
-Figure 4. Also need to statistically evaluated the differences.. Moreover Mean±SE ..But I have not seen any such value
We sincerely appreciate the valuable comments. We have revised this part according to the reviewer's suggestion.
Discussion
The discussion in written in good language but it’s not coherent with results. The result revealed the temporal changes of nutrients and chemical parameters. In the methodology physical factors are included but, in result the physical factors i.e. pH, Electrical conductivity is discussed, however, temperature data are missing and in the discussion, the changes in nutrients were linked with temperature. This need to be clarified, if temperature variations are considered for season than it needs to be evaluated. Some other minor corrections are as follow
-The section need to be checked for spelling and grammar mistakes
We sincerely appreciate the valuable comments. We have revised this part according to the reviewer's suggestion. We invited professional groups to help us improve our grammar and language skills.
Conclusion
The conclusion seems like summery of the results, it don’t provide further guidelines for study. If possible provide assessment about the outcomes and uses of the current study for future and also suggestion recommendation based on current findings.
We sincerely appreciate the valuable comments. We have revised this part according to the reviewer's suggestion.
Round 2
Reviewer 2 Report
The authors has substantially revised the manuscript and can be accepted in the current form.
Thank you